# Horses’ Tactile Reactivity Differs According to the Type of Work: The Example of Equine-Assisted Intervention

**DOI:** 10.3390/vetsci10020130

**Published:** 2023-02-07

**Authors:** Céline Rochais, Noémie Lerch, Léa Gueguen, Margaux Schmidlin, Ombeline Bonamy, Marine Grandgeorge, Martine Hausberger

**Affiliations:** CNRS, EthoS Department of Éthologie animale et humaine—UMR 6552, University Rennes, Normandie University, Station Biologique, 35380 Paimpont, France

**Keywords:** animal-assisted interventions, *Equus caballus*, horse-human relationship, perception, von Frey filaments

## Abstract

**Simple Summary:**

How animals perceive animal-assisted interventions (AAI) has been of concern lately, especially as these activities involve many tactile stimulations. It has been shown that tactile reactivity in horses could vary greatly between individuals and could depend upon a variety of factors. Repeated tactile stimulations, according to the associated valence, may lead to lower or higher reactivity. Here we hypothesized that the numerous tactile actions, sometimes with atypical gestures, during AAI may lead to tactile sensitization. In order to test this hypothesis, we tested, with von Frey filaments, the tactile reactivity of 60 horses involved in equine-assisted interventions (EAI), conventional riding school (RS) or mixed activities (EAI-RS). The results indicate that EAI horses showed a higher tactile reactivity than EAI-RS and RS horses, showing a higher number of reactions and a higher reactivity towards thin filaments. These differences could be related to human actions during EAI, as observations of brushing sequences by participants with or without mental and/or developmental disorders revealed differences in the distribution and modalities of tactile actions: participants diagnosed with such disorders brushed more the hindquarters and showed more fragmented actions. These results call for attention towards procedures during EAI and for promoting appropriate tactile actions from participants.

**Abstract:**

Tactile perception in humans varies between individuals and could depend on extrinsic factors such as working activity. In animals, there is no study relating the influence of animals’ work and their tactile reactivity *per se*. We investigated horses’ tactile reactivity using von Frey filament in different body areas and compared horses working only in equine-assisted interventions (EAI), in riding school (RS) lessons, and in both activities (EAI-RS). We further compared tactile actions by people with or without mental and/or developmental disorders during brushing sessions. The results indicated that EAI horses showed higher tactile reactivity compared to EAI-RS and RS horses, both in terms of number of reactions overall, and especially when the test involved thin filaments. All horses showed high tactile reactivity when tested on the stifle, and this was particularly true for EAI horses. These differences could be related to humans’ actions, as participants diagnosed with disorders brushed more the hindquarters and showed more fragmented actions. This study opens new lines of thought on the influence of EAI working activity on horses’ tactile reactivity, and hence, on horses’ sensory perception. Tactile reactivity outside work, may be directly (*via* tactile stimulations) or indirectly (*via* the welfare state), influenced by working conditions.

## 1. Introduction

Sensory perception contributes to the process by which mental representations are built from environmental stimulation [1]. Most behaviors that are at least partially triggered by external stimuli rely on perception. For example, mate choice necessitates first seeing, hearing or smelling a conspecific and recognizing it as a potential mate [1]. Sensory perception involves different modalities such as vision, olfaction, audition, gustation, and somesthesia. Somesthesia includes the perception of temperature, pain, itch, and tactile inputs such as the form and texture of objects. 

Tactile sensitivity, defined as the threshold differences in perception of tactile stimuli, has been especially studied in humans [2]. It is well established that tactile perception, and still more so, tactile reactivity, defined as differences in responsiveness, (i.e., perceiving a stimulus and responding to it with a particular behaviour), vary between individuals and depend on intrinsic factors such as age, sex (e.g., age [3]), and to some extent temperament [4]. Individual variations in tactile perception can also depend on extrinsic factors, which may induce either habituation or sensitization. For instance, it has been hypothesized that the higher tactile reactivity observed in premature human infants may be due to the repeated exposure to invasive procedures such as repeated venipunctures in neonatal care unit [5,6]. In adult humans, experience at work or practice in a specific domain related to the frequent use of the hands can contribute to a perceptual enhancement [7,8]. For example, musicians, including string instrumentalists and pianists, have been shown to perform better in tactile discrimination tasks compared to other non-musicians’ subjects [7].

Tactile experience could also be associated with particular affective valences, and hence, particular emotional memories, which can shape behavioural responses towards further experiences (e.g., approach or withdrawal) [9]. For instance, humans with low touch exposure evaluated the pleasantness of touch (gentle forearm stroking) differently from controls who received touch regularly [10]. Furthermore, tactile experience can be influenced by previous interactions with a conspecific and the experience of touch can switch from pleasure to displeasure if the perceived intentions or the identity of the emitter does not match the preferences of the recipient [11].

Human-domestic animal interactions involve many tactile stimulations. Tactile interactions may lead to contradictory reactions and sometimes long-term memories in the animals involved that vary according to the species, type of tactile stimuli, amount of restraint, or body area touched (e.g., neonatal handling, stroking style [12,13,14,15,16]). Specific activities such as animal-assisted interventions (AAI) involve many tactile interactions and have become increasingly popular [17]. AAI are defined as ”goal oriented and structured interventions that intentionally include or incorporate animals in health, education and human services (e.g., social work) for the purpose of therapeutic gains in humans” [18]. The question of how animals perceive tactile contacts during AAI in particular has been a concern in several studies lately [19,20]. These activities generally involve tactile actions, from soft touch to stroking, grooming or more invasive contacts (e.g., physical constraints related to human participants disabilities [21]), that may be potentially inappropriate, hence aversively perceived by the animals [22]. For instance, dogs have been shown to display increased redirected behaviors and displacement activities (reflecting discomfort) if an unfamiliar person petted them at the head, muzzle, paw, or tail but not if they were petted at the chest or neck [23]. All these results suggest that tactile human actions may lead to emotional memories in domestic animals, which may further differ according to the specific body areas touched. However, there is, to our knowledge, no study relating the influence of animals’ types of activities with humans and their tactile reactivity *per se*.

Domestic horses constitute a very interesting model as there are clear individual differences in tactile reactivity that depend both on intrinsic and extrinsic factors [24]. Moreover, despite numerous tactile contacts with humans and materials during work, no specific study has been conducted on possible effects of working conditions on tactile reactivity [24]. Horses are particularly sensitive to skin stimulation, with the panniculus reflex, i.e., skin tremor, being triggered by a fly landing on the skin [25], and respond to the application of a standardized stimulus as light as 0.008 g [26]. Furthermore, horses do not express a large amount of tactile contact during social interaction and mutual grooming accounts for only 2–3% of a horses’ time budget [27] and contacts are limited to specific body parts [28]. However, in domestic situations, human-horse interactions involve many “strong” tactile contacts, from patting, grooming, tacking and untacking, to hand and leg contact during riding. Physical contacts can vary in quantity and quality depending on the type of work/activity (and human’s training), leading to potential different emotional memories (e.g., [12]).

In a recent literature review on horses’ tactile reactivity tested with the same calibrated tool (von Frey filaments), Gueguen et al. [24] first showed that discrepancies between study procedures prevented definite conclusions on many possible factors of influence on tactile reactivity. Second, both intrinsic (e.g., sex, age, type of equid, welfare state) and extrinsic (e.g., living conditions) conditions were likely to have an influence on horses’ tactile reactivity. Whether working conditions may have an influence remains to be studied but most training procedures involved in “classical” riding practices aim to reduce reactions to tactile stimuli (e.g., imprint training [29]) and use repeated tactile stimulations that may lead to either habituation or sensitization, as in humans (e.g., [30]). One study found that sport ponies were more reactive than recreational ponies independent of breed, suggesting a potential influence of the type of work, but other factors, such as the usual living conditions may also have differed [31]. Similarly, Vidament et al. [32] found that horses ridden by lower level riders had a lower tactile reactivity than those ridden by experienced riders, but there was no indication on other possible factors of influence. Horses’ emotional, welfare and cognitive states outside working sessions are highly influenced by working conditions [33,34,35,36,37,38,39]. Therefore, it is likely that tactile reactivity outside work, may be directly (*via* tactile stimulations) or indirectly (*via* the welfare state), influenced by working conditions.

In order to test this hypothesis, we compared the tactile reactivity of horses involved in equine-assisted interventions (EAI) to that of horses involved in conventional riding school activities. Horses are one of the most frequently used species in AAI (*e.g.,* for children with autism spectrum disorders [40]) where they are confronted to participants with different disorders (e.g., psychological, physical, social disorders) that may present unusual (for the horse) movements or emotional reactions [21,41,42]. EAI generally include a large part devoted to tactile contact (e.g., grooming, stroking in all cases, and riding or harnessing for some of them) [21]. How animals perceive these contacts is an important question, especially as there are some hints that EAI horses show less interest towards humans or even more avoidance in human-horse relationship tests compared to riding school or sport horses [43,44]. These previous results could be related to a negative perception of human actions, including tactile actions. Repeated actions perceived negatively could alter human–horse relationship especially as horses are able to generalise the relationship they have with their familiar humans (e.g., rider) to an unknown human [45,46]. Furthermore, a previous study showed that tactile interaction during overall classical grooming procedures induce negative immediate reactions in riding school horses [47]. We predicted that because the atypical gestures and the high amount of tactile contact during EAI sessions may be perceived negatively and more invasive by EAI horses, they may show a higher tactile reactivity outside work.

We tested, using von Frey filaments as a consensual methodological tool to measure tactile reactivity, EAI and non-EAI horses, which, in order to diminish possible biases related to other factors, were in the same overall living conditions and tested in the same conditions (familiar location, same time of year, same weather) (see [24]). Finally, in order to have an idea of potential factors involved in tactile reactivity differences, we investigated spontaneous brushing procedures during grooming sessions of horses by adult humans presenting mental and/or developmental (e.g., autism) disorders and adults without any disorder (physical or mental) diagnosed.

## 2. Material and Methods

### 2.1. Ethical Statement

The experiments were carried out between February and May 2019 in accordance with the Directive 2010/63/UE of the European Parliament and the Council on the protection of animals used for scientific purposes. They complied with the current French laws related to animal experimentation (decree No. 2013 ± 118 of 1 February 2013) and its five implementation orders (JO 7 February 2013, integrated into the Rural Code and the Code of maritime fishing under No. R. 214 ± 87 to No. R. 214–137). The experiments performed in this study were not within the scope of application of the European directive; thus, in accordance with this directive and the current French and Irish laws, the following experiments did not require us to request authorisation. These experiments involved only behavioural observations and non-invasive interactions with the horses. The horses used in this research were not research animals. Animal husbandry and care were under the management of the riding school staff. The riding school managers gave the authors their informed consent for this study.

The present research involving human participants was in accordance with the directive No. 2004–801 of 6 August 2004; law No. 78–17 of 6 January 1978, JORF of 30 May 2019 and with the Declaration of Helsinki (6th revision). The European GDPR (General Data Protection Regulation) was fully applied, and all participants were informed of the subject of the study, of the protocol for processing personal data, video recordings and of their rights of withdrawal. For all participant, access to experiments was conditional on an explicit, free, informed and written consent to participate, obtained by signing themselves or signature from their legal guardians. Data were all collected and analyzed anonymously.

### 2.2. Study 1: Horses’ Tactile Reactivity in EAI and Non EAI Horses

#### 2.2.1. Methods

##### Study Sites

This study involved 2 different riding centres, centre 1 located in North-East part of France where observations were performed between February and March 2019 and centre 2 located on the east coast of Ireland where experiments were performed between April and May 2019. Climate and temperature were the same in both centres (mean_Nancy_ = 6.1 °C; mean_Dublin_ = 6.3 °C) at the time of testing. The proportion of each activity was between 7% and 86% of equine assisted activities (EAI), mean ± SE = 40.5 ± 5.1% of the total amount of working hours per week. EAI activities consisted mostly in grooming, groundwork, lunging, and riding. It involved persons with disabilities such as motor disabilities, visual, hearing, or cognitive impairments, crippling diseases, or other health diseases linked to psychosocial risks or social problems (e.g., jail, dropping out of school). Information on management practices were obtained through a questionnaire filled by riding centre managers. We found no effect of the riding centre on tactile reactivity (see results), thus individual data of horses were pooled between centres.

##### Horse Subjects

This study included 60 horses (32 mares, 28 geldings, aged 4 to 27 years old, mean ± SE = 14.06 ± 0.70, Table 1) that had lived for at least one year within the center and were tested in one of the 2 different riding centres. There were 23 pony-type equids (≤148 cm at withers) and 37 horse-type equids (>148 cm at withers) (International Federation for Equestrian Sport) from various breeds but mostly unregistered (Table 1). At the time of the study, they had been involved in the same working activity for at least one year. Overall, 6 equids worked only in equine-assisted interventions (EAIs), 14 in ‘conventional’ riding school lessons (RS), and 40 in both activities (EAI-RS) (Table 1). Horses were in the same overall living conditions whatever their working activity. They were mostly housed indoors either in single (63%) or collective stalls (37%) with water provided *ad libitum* through automatic drinkers. Daily turn out was ensured in both facilities.

##### Tactile Reactivity Test

Tactile reactivity was estimated using von Frey filaments (Stoelting, IL, USA). These filaments consist of a hard-plastic body extended by nylon threads of variable thickness and are calibrated to apply a specific force on the skin, making it possible to obtain standardized measures [24]. When applying the tip of the filament perpendicularly against the skin, the application force increases as long as the experimenter continues the pressure, until the filament flexes and the filament is then removed. The procedure was the same as in Gueguen et al. [24]: we applied the von Frey filaments at 3 different body areas on both sides: the chest, the base of the withers and the stifle as skin sensitivity varies across a horse’s body depending on the distribution of sensory nerve receptors [48]. We also decided to apply the filaments in different body area in order to explore the potential impact of work [24]. We used 4 different filament sizes (0.008 g, 0.02 g, 1 g, and 300 g). Previous studies showed that the number of horses reacting to von Frey filaments increased significantly between 0.008 g and 8 g, whereas above 8 g, fewer horses reacted [26]. The tactile reactivity test was performed in two sessions at least 30 min apart. For each session, the order of the body sides, of the body areas, and of the filament sizes was random. Thus, for each session, two filaments were applied, first on one side for the three areas and then on the other side for the same three areas. Overall, 24 tests of tactile reactivity were performed per horse (3 body areas × 2 sides × 4 filaments sizes). For each test, the horses’ responses to each von Frey stimulation were coded in binary form (tremor = 1/no tremor = 0). Since there was no variation in total reactivity according to horses’ body side (Wilcoxon test: EAI horses: V = 4, P = 0.416; EAI-RS: V = 176.5, P = 0.370; RS: V = 11, P = 0.100), both sides were summed up for further analyses. Tactile reactivity was tested by the same experimenter (NL, unknown to the horses at the time of testing) in its usual environment (home stall).

##### Data and Statistical Analyses

All statistics were performed using R 3.6.2 software (The R foundation for statistical computing, http://www.R-project.org/, accessed on 1 January 2020, Vienna, Austria) [49]. The significance level was set at 0.05. Descriptive statistics are reported as means and standard error of mean (SEM). As the data were not normally distributed, we used non-parametric statistical tests [50].

The following data were used to characterize horses’ tactile reactivity:(1)Total reactivity: sum of all responses, up to a maximum of 24 (4 filaments tested on both sides on 3 areas of the body).(2)The reactivity by filament size: the sum of the responses for each size of filaments tested. Since data were similar for thin (0.008 g and 0.002 g) or thick (1 g and 300 g) filaments, respectively, we used these categories for statistical analyses.(3)Reactivity per side of the body: the sum of the responses on the left and right sides of the body (areas and filaments pooled), respectively.(4)Reactivity by body area: the sum of the responses recorded for each body area tested (sides and filaments pooled).

Kruskal-Wallis tests followed by post-hoc pairwise Mann-Whitney U tests were used to compare each variable (i.e., total reactivity, reactivity by filament size (thin *versus* thick), reactivity per body area) according to horses’ activity (RS vs. EAI vs. EAI-RS). Within horses’ activity groups, Wilcoxon tests were used to compare reactivity per side and filament range.

Mann-Whitney U tests were used to assess the influence of intrinsic factors such as age (between 0 and 15 y.o *versus* ≥ 15 y.o, which is commonly used in previous studies [43,51] and as horses over 20 years old would have a lower tactile reactivity [25]), sex (mares *versus* geldings), and type of horses (ponies *versus* horses), as well as the influence of extrinsic factors such as the riding centre.

#### 2.2.2. Results

The total tactile reactivity (i.e., total number of reactions for all tests and including all body areas and filament range) showed high individual variations ranging from 0 to 24 (9.93 ± 0.81). More precisely, there were clear differences according to the horses’ usual working activity: EAI horses showed a higher reactivity (14.5 ± 2.0) compared to EAI-RS horses (9.8 ± 1.0) (N_(EAI)_ = 6; N_(EAI-RS)_ = 40, Mann-Whitney U-test: W = 17, *p* = 0.043), and RS horses (8.3 ± 0.6) (N_(EAI)_ = 6; N_(RS)_ = 14, W = 48, *p* = 0.019) (Figure 1). No difference was found, however, between EAI-RS and RS horses (N_(EAI-RS)_ = 40; N_(RS)_ = 14, W = 225, *p* = 0.281).

Furthermore, differences appeared according to filament size. Thus, whereas EAI horses did not show any significant difference according to filament size (thin filaments: 7.2 ± 0.9; thick filaments: 7.3 ± 0.5; V = 6, *p* = 0.787), EAI-RS and RS horses showed a lower reactivity towards thin filaments (0.008g and 0.02g; EAI-RS: 4.1 ± 0.5; RS: 3.2 ± 1.1) than towards thick filaments (1 g and 300 g; EAI-RS: 5.8 ± 0.5; RS: 5.1 ± 1.1; Wilcoxon test: EAI-RS: V = 67.5, *p* = 0.0007; RS: V = 8, *p* = 0.028). This was confirmed when comparing the response of EAI horses and other horses: EAI horses showed a higher tactile reactivity when tested with thin filaments (7.2 ± 0.9) than EAI-RS (4.1 ± 0.5, W = 17, *p* = 0.043) and RS horses (3.2 ± 1.1, W = 46.5, *p* = 0.017) whereas there was no such difference with the thick filaments (Kruskal-Wallis tests: *X*^2^ = 2.2, df = 2, *p* = 0.334). No significant difference was found between EAI-RS and RS horses (i.e., thin filament, W = 215, *p* = 0.199).

Tactile reactivity varied also according to body area. It was higher overall on the stifle (EAI: 6.3 ± 0.6; EAI-RS: 4.4 ± 0.3; RS: 3.4 ± 0.7) than on the chest (EAI: 3.3 ± 0.7; EAI-RS: 2.1 ± 0.4; RS: 1.7 ± 0.7; Wilcoxon test: EAI: V = 0, *p* = 0.058; EAI-RS: V = 575.5, *p* < 0.001; RS: V = 52.5, *p* = 0.012). However, differences appeared according to the horses’ usual working activity. EAI-RS and RS horses showed higher tactile reactivity when tested on the withers (EAI-RS: 3.4 ± 0.4; RS: 3.2 ± 0.8) compared to the chest (EAI-RS: 2.1 ± 0.4, V = 289.5, *p* = 0.004; RS: 1.7 ± 0.7, V = 63, *p* = 0.008) but no such significant difference was found for EAI horses (wither: 4.8 ± 0.6; chest: 3.3 ± 0.7; V = 2, *p* = 0.361). EAI-RS horses only showed a higher tactile reactivity when tested on the stifle compared to the wither (EAI: V = 5, *p* = 0.292; EAI-RS: V = 129, *p* = 0.007; RS: V = 37, *p* = 0.749). Moreover, EAI horses showed higher tactile reactivity when tested on the stifle compared to EAI-RS (W = 56, *p* = 0.036) and RS horses (W = 14, *p* = 0.022), whereas no significant difference was found between EAI-RS and RS horses (W = 215.5, *p* = 0.202). No statistical difference was found according to horses’ activity when they were tested on the chest (*X*^2^ = 4.3, df = 2, *p* = 0.116) and the wither (*X*^2^ = 2.1, df = 2, *p* = 0.350) (Figure 2).

There was no significant influence of the riding centre on the total tactile reactivity (centre 1: 9.8 ± 1.2; centre 2: 10.1 ± 1.1; W = 446, *p* = 1), nor of equids’ age (≤15 y.o: 10.1 ± 1.1; ≥15 y.o: 9.6 ± 1.1; W = 457, *p* = 0.637), type (horses: 8.5 ± 1.0; ponies: 11.1 ± 1.2; W = 347, *p* = 0.158), and sex (mares: 8.7 ± 1.0; geldings: 11.4 ± 1.0; W = 330.5, *p* = 0.082).

#### 2.2.3. Conclusion of Study 1

The results of this study revealed that EAI horses showed higher tactile reactivity, both in terms of number of reactions to von Frey filaments overall but especially when the test involved thin filaments, suggesting a possible sensitization towards tactile stimulation. All horses showed higher tactile reactivity when tested on the stifle, but this was particularly true for EAI horses. EAI-RS and RS, but not EAI horses showed higher tactile reactivity when tested on the withers than on the chest. The results suggest a relationship between working activities and horses’ tactile reactivity tested outside working sessions.

### 2.3. Study 2: Comparison of Grooming Procedure between Persons Diagnosed with or without Mental and/or Developmental Disorders

#### 2.3.1. Methods

##### Subjects


**Human Participants**


In this case, 46 adults, 18 to 45 y.o, were recruited in riding centres dedicated to EAI: 17 (7 women, 10 men, age: ±SE = 25 ± 2.3 y.o) were diagnosed with mental and/or developmental (e.g., autism) disorders and 29 (18 women, 11 men, age: ±SE = 27 ± 1.1 y.o) were without any mental, developmental or physical disorder (considered as neurotypical adults). Diagnosis had been made according to the DSM-IV [52] and ICD-10 [53] criteria (category F80–F89: mental, psychic or social disability) by specialists. This information was transmitted by participants themselves or caregivers. Most participants were practising horse riding (47%) on a weekly basis.


**Horses’ Population**


A total of 29 horses living in five different centres (X = 5 horses per centre) were involved in the study. They corresponded to three categories:–Nine horses (6 mares, 3 geldings, age: ±SE = 13.4 ± 1.4 y.o) that had been involved only in EAI for at least one year. They lived in groups, in pasture with hay and water *ad libitum*.–Nine horses (7 mares, 2 geldings, age: ±SE = 14.4 ± 1.3 y.o) that had worked in conventional riding centre lessons for at least one year and lived in straw bedded, 3 × 3 m^2^ individual stalls with water provided *ad libitum* and hay and commercial pellets provided twice a day.–Eleven leisure horses (6 mares, 2 geldings, 3 stallions age: ±SE = 11.4 ± 2.2 y.o) that were occasionally ridden outdoors and lived in groups in pastures with water *ad libitum* and hay during winter.

##### Grooming Session

Human participants’ spontaneous behaviours were videorecorded (Sony HDR-XR105^®^ version 1, disposed on tripods on each side of the horse) during a classical grooming session. Each horse was tested with one person with or without disorder. Thus, each horse was tested twice. Grooming sessions were performed in horses’ familiar area. The horse was held slightly with a halter and loose lead rope by an unfamiliar experimenter facing it and standing still, gazing towards the ground, with no interaction with the horse except in case of horses’ movement. The horse was put back in its place in case of a single movement (e.g., one step forward) or the session was stopped in case of two or more movements. A second experimenter (familiar for participants) accompanied the human participant walking towards the horse and gave the instruction to brush the horse with a soft brush for one minute on each body side before leaving the testing area. Participants were told they were free to brush the horse as they wished. Horses’ body side order was chosen randomly.

##### Data Recording and Analyses

Participants’ behaviours during the grooming session were analysed from the video footages. In order to have a fine description of the time spent grooming each body area, data analysis was performed using continuous focal sampling [54]) by the same experimenter (MS). Given the shortness of the grooming sessions and related low data sampling, we analysed further the data by considering two main “classical” areas [55]: (1) forehand (head, mane, neck, wither, shoulder, forelimbs); (2) hindquarters (back, flank, croup, hind limbs). The number of grooming stops (i.e., breaking contact between the brush and the horse’s body) was measured and a brushing fragmentation index was calculated: number of brushing stops divided by the total duration of brushing time during the test.

All statistics were performed using R 3.6.2 software [49]. The significance level was set at 0.05. Descriptive statistics are reported as means and standard error of the mean. As data were not normally distributed, we used non-parametric statistical tests [50]. Mann-Whitney U tests were used to compare the time spent grooming each body area and the brushing fragmentation between participants with or without disorder. Friedman and Wilcoxon signed rank tests were used to compare the time spent grooming the two main areas (forehand and hindquarters).

#### 2.3.2. Results

Participants without disorder brushed a larger number of body areas (X_without disorder_: 6.1 ± 0.3; X_with disorder_: 4.6 ± 0.5; W = 137, *p* = 0.012) and spent more time brushing horses’ forehand (51.2 ± 5.3 s; 32.1 ± 5.7; W = 135, *p* = 0.010) compared to participants with disorder. They also showed no brushing preference for any of the body areas (V = 193, *p* = 0.601) contrarily to the participants with disorder who spent more time brushing the horses’ hindquarters than their forehand (V = 28, *p* = 0.020) (Figure 3). Participants with disorder showed more fragmented brushing actions (fragmentation index: 0.021 ± 0.006) compared to participants without disorders (0.005 ± 0.001; W = 347, *p* = 0.017).

#### 2.3.3. Conclusion of Study 2

Observations of grooming sequences by people with mental and/or developmental disorders or not revealed differences in the distribution of tactile actions on the horses’ body. People with mental and/or developmental disorders showed more actions directed at hindquarters and more fragmented actions.

## 3. General Discussion

The present study explored the reactions of horses involved in equine-assisted interventions, conventional riding school lessons or mixed activities, to a standardized tactile test outside work. The results revealed that EAI horses showed higher tactile reactivity, both in terms of number of reactions to von Frey filaments overall but especially when the test involved thin filaments. All horses showed higher tactile reactivity when tested on the stifle, but this was particularly true for EAI horses. EAI-RS and RS, but not EAI horses showed higher tactile reactivity when tested on the withers than on the chest. Interestingly, observations of brushing sequences by participants with or without mental and/or developmental disorders revealed differences in the distribution of tactile actions on the horses’ body, with more actions directed at hindquarters in the latter, and also more fragmented actions, that might well be perceived as rougher.

### 3.1. Higher Tactile Reactivity in EAI Horses

In the present study, no difference in horses’ tactile reactivity appeared according to age, sex, type of horse (horse *versus* pony) or riding center. However, there were clear relationships between working activities and horses’ tactile reactivity tested outside working sessions. EAI horses showed a higher tactile reactivity than EAI-RS and RS horses. One could have expected that equids chosen for EAI activities would have shown a lower tactile reactivity as they are expected to remain calm whatever the person’s actions [44,56]. However, the studies aiming at comparing EAI and non-EAI horses’ temperament have failed up to now to show differences [56,57,58]. Thus, the present results suggest more a potential input of environmental influences on tactile reactivity.

In a recent pilot study on recreational horses with a low or no amount of riding, the average number of reactions in response to von Frey filaments was 5.75 [24], whereas in the present study it was much higher (9.93) especially in EAI horses (14.5). These results highlight the potential impact of working activity. This is particularly confirmed by differences in tactile reactivity according to the body area tested. In Gueguen et al. study [24] unridden horses showed higher reactivity on the withers whereas in the present study, horses showed higher reactivity on the stifle. These results suggest a possible sensitization towards tactile stimulation related to working activities, and still more so where EAI are involved.

Amongst each working condition (i.e., EAI/EAI-RS/RS), all horses are subjected to daily tactile stimulations from grooming, to handling, saddling, hand and leg contact during riding. However, EAI sessions are more often characterized by grooming or “on foot” exercises [21], or involve specific actions such as lying on horses’ croup or double riding (rider and caretaker on horses’ back) with heavy equipment that potentially lead to particular and numerous tactile stimulations. This could mean that, if that part of the activity is influential, horses with mixed activities (both EAI and RS lessons) could show intermediate reactions. Interestingly though, EAI horses differed from both other work categories of horses, while RS and EAI-RS did not differ significantly. Differences involved the overall tactile reactivity, the reactivity according to the filament range and even the body area tested. Only EAI horses showed the same high reactivity when tested with thin and thick filaments highlighting again a possible sensitization towards tactile stimulation. For instance, it has been shown that preterm infants tested at term-age showed a higher tactile sensitivity compared to full-term neonates suggesting an influence of repeated tactile stimulations at that time (e.g., repeated venipunctures) which may have led to sensitization [6]. Young horses at the beginning of their training without much tactile stimulation yet as well as recreational horses with a low or no amount of riding react less to thin compared to thick filaments [26,31]. These results further suggest a possible role of working activities in the observed higher horses’ tactile reactivity as “non-working” activity could be related to the absence of sensitization. The same parallel is found in humans where the influence of humans’ experience at work related to the frequent use of the hands can contribute to a perceptual enhancement and higher tactile sensitivity [7,8].

### 3.2. Differences in Tactile Reactivity between Body Areas and According to the Working Activity

The hypothesis of tactile sensitization in EAI horses related to working conditions is also partially confirmed by tactile reactivity differences according to the body area tested. Overall, all horses showed lower reactivity when tested on the chest compared to the withers and stifle, in accordance with Fureix et al. [59] and Gueguen et al. [24]’s results. Horses’ reaction threshold towards nociceptive stimulations have been shown to decrease along the spine [60]. In addition, the distribution of sensory nerve receptors differs between tested areas, with a higher number at the nose, neck, withers, shoulders, the flank, and back of the pastern [48]. This suggests differences in tactile sensitivity over the body, but is not necessarily predictive of tactile reactivity. Thus, in the present study, differences in tactile reactivity appeared between body areas according to working activity. EAI horses showed the same high tactile reactivity over all body areas but differed from EAI-RS and RS horses as they reacted more on the stifle whereas EAI-RS and RS horses on the contrary showed the same pattern with a higher tactile reactivity on the withers compared to the chest. Variation in tactile reactivity have been found in donkeys used in EAI and tested with von Frey filament, where the most responsive areas were the withers, back, forelimbs, ribs, and stomach, while neck, rump, and forehead showed the lowest reactivity [20]. Furthermore, van Iwaarden et al. [30] showed an impact of the saddle position on “athletic” horses’ muscle anatomy. They suggested a possible failure to habituate to pressure of the saddle that may result in continuous stimulation of the panniculus reflex, which could play a role in persistent sensitivity to girth pressure in some horses [30]. Of course, saddling induces repeated tactile stimulations around the withers and girth body areas in equids used for riding which could influence tactile reactivity. It is interesting to find higher tactile reactivity at the stifle area in EAI horses, but it might be related to the main difference between EAI and RS equids, that is, the higher number of tactile actions in EAI either “on foot” or when ridden [21].

Tactile experience could also be associated with particular emotional valence, and hence, particular emotional memory. In foals, tactile stimulations, according to their quality, amount or body location, may induce short term reactions [12,13,14,61], but also create long-term emotional memories [14,61]. In piglets, only a few days of negative tactile contact (being captured and lifted by a human handler) lead to increased fear reactions towards humans 5 weeks later [62]. In humans, the pleasantness value of touch depends on positive or negative previous experiences with the emitter [9]. Thus, the combination of the influence of atypical gestures, contact quality and numerous tactile stimulations in potentially sensitive body areas during EAI may have an impact on horses’ emotional memory towards tactile stimulation.

### 3.3. Differences in Brushing Location and Modalities between People with or without Mental and/or Developmental Disorders in Grooming EAI Sessions

Observations of brushing procedures during grooming revealed that participants with mental and/or developmental disorders spend more time at the horses’ hindquarters than forehand, contrarily to participants without disabilities. Moreover, they have more fragmented actions that may be perceived as less smooth by horses. For instance, in humans, higher pleasantness ratings were found during stroking touch from 1 to 10 cm/s compared to faster stroking velocities in adults [63]. Further studies measuring brushing pressure and velocity may be of great interest. There is no clear explanation at that stage on why those beneficiaries all privileged the hind body part of horses, but this may explain for some part at least that EAI horses, if they perceived these actions as more invasive, develop higher responsiveness, if not avoidance. Lerch et al. [43] and Brubaker et al. [64] have shown that EAI horses tend to be less interactive when confronted to humans than RS horses, while Mendonça et al. [65] describe them even as more “apprehensive” of human contact.

Previous studies on RS horses have shown that the “standard method” of grooming horses induces avoidance and threatening behaviors suggesting that these actions caused discomfort [47,66]. The authors suggested to improve practice by grooming supposedly preferred zones such as the withers, as a preferred area for allogrooming between horses and as human scratching the wither area results in a decrease in heart rate [28]. Unfortunately, the authors do not indicate whether grooming some particular body areas was more associated with discomfort behaviors and whether there were less of them at withers. Furthermore, earlier findings showed that not all horses perceived tactile stimulation at the wither as positive [38,67]. It is noteworthy that positive or negative perception of tactile stimulation at the wither could depend on environmental factors such as the season. For instance, unridden horses tested with the same von Frey filament test, showed higher tactile reactivity in summer than in winter [24]. Several hypotheses could explain these results, such as the impact of thicker horses’ coat in winter [68] and/or an accumulation of fat tissue over the entire body surface [69]. Tactile stimulation at the wither could be more pleasant during moulting season or enhanced for hygienic purpose (e.g., presence of lice) [70]. Thus, particular horses’ body areas may be more or less appropriate for tactile stimulation from a horse’s point of view [71].

The body area stimulated may be of importance as the horses’ reactions may depend on a combination of (i) specific preferential parts of the body related to intra-specific social behaviour, (ii) inter-individual differences in preferred body area for tactile stimulation and (iii) experience with human actions. Some privileged grooming/tactile contact zones have been described in other species. Thus, heifers show positive reactions (i.e., seeking human contact) when stroked at the dorsal or ventral neck, cheek and jaw but not at the muzzle, forehead, ear, poll and back [16]. Piglets that had received gentle contacts by a human (i.e., softly stroking along the body) for 3 weeks, showed a decrease in heart rate variability (supposed to increase in positive situations) when scratched at the rear contrarily to control (no contact with humans) piglets [15]. The authors suggested that being scratched at the rear was perceived as positive by control piglets, but not by previously handled piglets. Similar results have been found in cats, where higher expression of negative behaviors towards humans (i.e., bite, cuff, tail flicking) was found when petted on the caudal region [72]. Dogs have been shown to display increased discomfort behaviors if they were petted at the head, muzzle, paw or tail but not if they were petted at the chest or neck [23]. In the same vein, young horses trained to accept usual handling procedures (i.e., being fitted with a halter and a surcingle, giving their feet, etc.) showed different reactions to a human approach according to the direction of approach [73]. The front, the left and right shoulders were easier to approach and right croup more difficult whereas left and right shoulders, as well as left flank were more difficult to touch in naïve, untrained horses [73]. However, when considering forehands *versus* hindquarters, the latter appeared less easy to approach (13/23 horses with negative reactions when approaching the hindquarters *versus* 2/23 negative reactions when approaching the forehands), showing that hindquarters are overall less privileged zones for contact [73]. This is in accordance with observation on social contact between horses, where positive approaches are described as more frontal (i.e., conspecifics approaches and investigations at the nose, [74]).

## 4. Conclusions

The results of this study open new lines of thought on the possible influence of EAI working activity on horses’ tactile reactivity, and hence, on horses’ sensory perception. In view of the results, we propose a direct impact of work on the tactile reactivity via a potential sensitization to repeated inappropriate tactile stimulations in non-preferred body areas. Observations of brushing during EAI sessions could confirm the potential impact of work as differences in the distribution and modalities of tactile actions on the horses’ body were found between people with or without mental/developmental disorders. Further studies are needed on larger samples and integrating the influence of other work aspects such as the percentage of time spent in EAI *versus* other working activity (i.e., classical riding school) and how tactile stimulations differ between EAI and other working activity either during grooming or riding. For instance, further investigations are required on the influence of equipment used during grooming and riding in EAI (e.g., “double” saddle may have an influence on horses’ back body area). Further studies should compare tactile actions between people with different psychological and/or physical disorders. These could help identifying which specific tactile actions cause discomfort and help to advise riding instructors to teach a better way for grooming horses. Associating sessions with positive (food) reinforcement could also lower the negative valence of some actions and promote horse’s motivation for human contact. Finally, further investigations integrating horses’ welfare indicators during grooming and riding in EAI should help understanding further potential links between expressions of compromised welfare and tactile reactivity.

## Figures and Tables

**Figure 1 vetsci-10-00130-f001:**
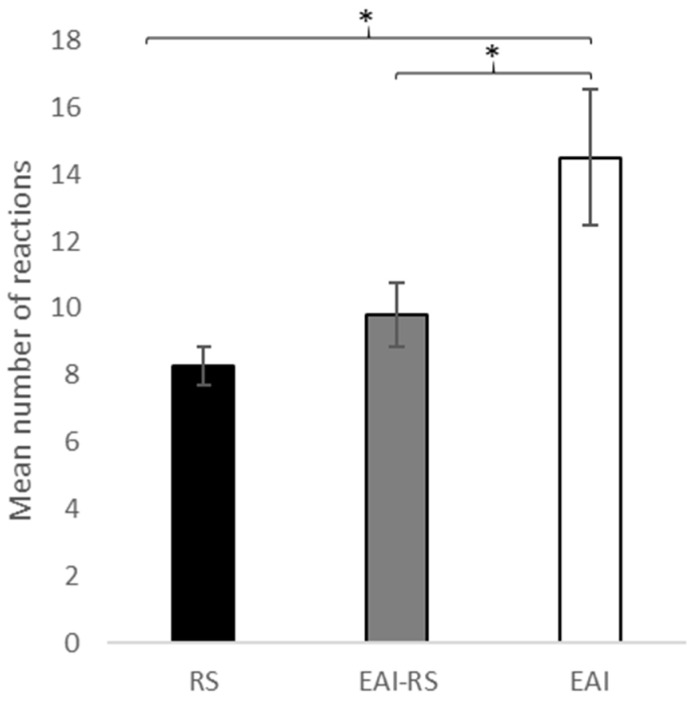
Mean number of total number of reactions (tremor = 1/no tremor = 0) to the tactile reactivity test using von Frey filaments. The data represent the sum of all responses (4 filaments tested on both sides on 3 areas of the body). RS = riding school lessons horses (N = 14), EAI = assisted-intervention horses (N = 6), EAI-RS = horses with mixed activity (N = 40). Kruskal-Wallis and Mann-Whitney U tests, * *p* < 0.05.

**Figure 2 vetsci-10-00130-f002:**
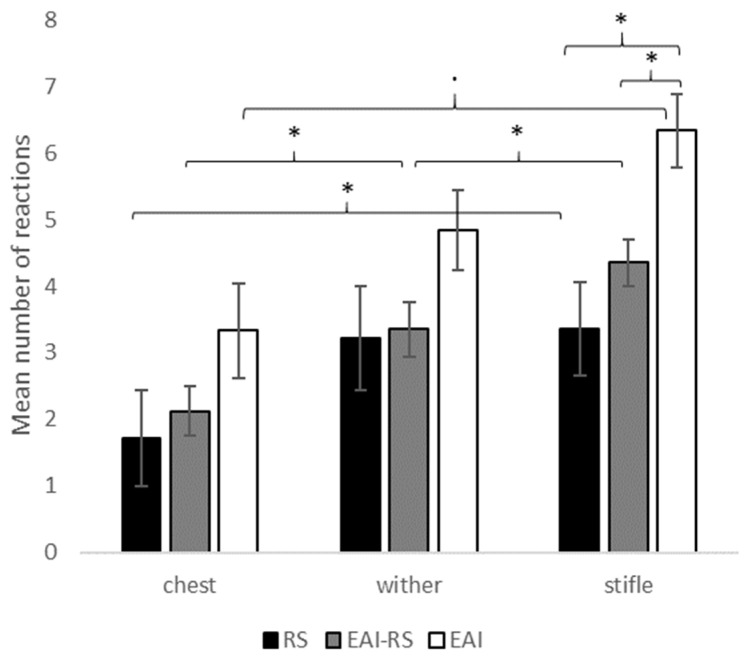
Mean number of reactions (tremor = 1/no tremor = 0) during tactile reactivity tests according to the body area tested (chest, wither, stifle). RS = riding school lessons horses (N = 14), EAI = assisted-intervention horses (N = 6), EAI-RS = horses with mixed activity (N = 40). Kruskal-Wallis and Mann-Whitney U tests between horses’ activity and Friedman and Wilcoxon tests between body areas, · *p* < 0.1, * *p* < 0.05.

**Figure 3 vetsci-10-00130-f003:**
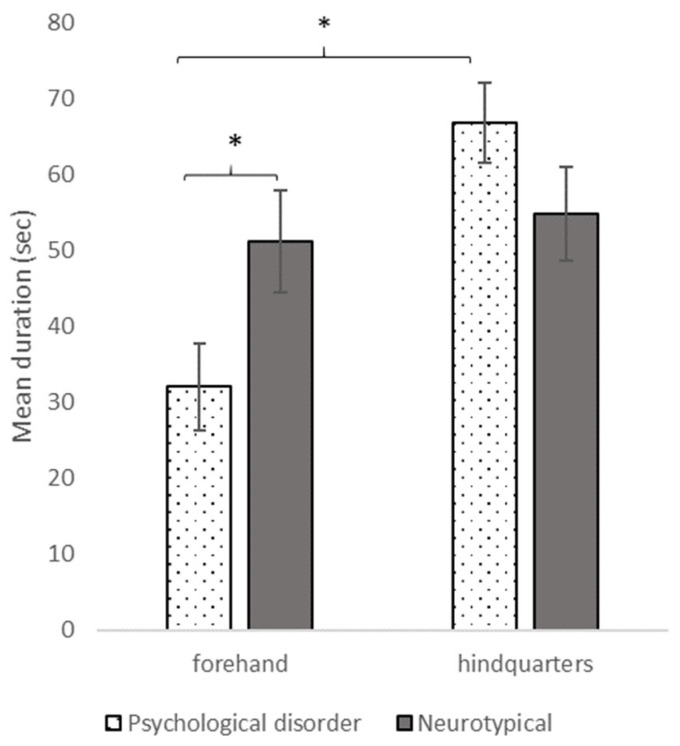
Mean duration (in seconds) of brushing different horses’ body area between participant with psychological disorder (N = 17) and without (N = 29). Mann-Whitney and Wilcoxon tests, * *p* < 0.05.

**Table 1 vetsci-10-00130-t001:** Subject characteristics and management practices in the two study riding centers.

		Centre 1	Centre 2
Equid type	Ponies	12	11
	Horses	15	22
Sex	Mares	20	12
	Geldings	7	21
Age (mean ± SE) Y.O		13.2 ± 1.2	14.8 ± 0.8
Riding activity ^1^	RS	2	12
	EAI	5	1
	EAI-RS	20	20

^1^ Note. RS = riding school lessons horses, EAI = assisted-intervention horses, EAI-RS = horses with mixed activity.

## Data Availability

The data generated and analysed during the current study are available from the corresponding author on reasonable request.

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
