# Peer review of "Horses’ Tactile Reactivity Differs According to the Type of Work: The Example of Equine-Assisted Intervention"

_vetsci, 2023, doi:10.3390/vetsci10020130_

Round 1
Reviewer 1 Report
This is a worthwhile paper. The topic is introduced well, outlining that the natural and domestic types of tactile contact are very different. The methods are clear. The results are well-written and thoughtfully discussed.
Please note that I am reviewing this paper due to my experience in equine welfare science and I have no expertise in EAI. Also, although I can follow and understand statistics, I have no expertise in assessing the validity of different statistical tests so have not commented on the stats elements of the paper - this will need to be checked by an appropriately qualified reviewer.
- Some small typos - e.g. Nord not North France on line 178, daily turnout was (not were, line 202), except not expect in line 346, reactions not reaction in line 509 - these are just some I noted.
- I was interested that the more fragmented brushing was thought to be rougher than less fragmented brushing - could it be perceived as being more ticklish/annoying? Indeed sometimes more pressure might be better tolerated than less pressure.
The study raises lots more questions for further research and is a valuable addition to the scientific literature considering the use of animals in human therapy. I have very minimal comments and recommend publication.
Author Response
This is a worthwhile paper. The topic is introduced well, outlining that the natural and domestic types of tactile contact are very different. The methods are clear. The results are well-written and thoughtfully discussed.
Thank you
Please note that I am reviewing this paper due to my experience in equine welfare science and I have no expertise in EAI. Also, although I can follow and understand statistics, I have no expertise in assessing the validity of different statistical tests so have not commented on the stats elements of the paper - this will need to be checked by an appropriately qualified reviewer.
- Some small typos - e.g. Nord not North France on line 178, daily turnout was (not were, line 202), except not expect in line 346, reactions not reaction in line 509 - these are just some I noted.
All typos have been corrected.
- I was interested that the more fragmented brushing was thought to be rougher than less fragmented brushing - could it be perceived as being more ticklish/annoying? Indeed sometimes more pressure might be better tolerated than less pressure.
The hypothesis is that fragmented contacts may be perceived negatively either through being more ticklish or surprising.
The study raises lots more questions for further research and is a valuable addition to the scientific literature considering the use of animals in human therapy. I have very minimal comments and recommend publication.
Thank you
Reviewer 2 Report
Overall: Exceptionally well-written and clear with regards to literature review, methods, results, and discussion of the results in the context of existing studies. Very well done!
Abstract: Well-written and covers the hypothesis, materials and methods, findings, and conclusions with concise clarity.
Line 28- a word is missing between “such” and “working”
Introduction: Overall, very thorough and well-written, covering all important topics related to the experiment.
Line 47 – 48: Consider mentioning that olfactory senses of environmental hormones are also involved in mate choice
Line 55-63: Consider adding in that differences in skin nerve receptors can also play a role.
Materials and Methods: Overall very good and thorough in the explanation of the experiment and design.
Line 193: Please clarify the “lived for at least one year”. Do you mean within the center or among familiar horses, etc?
General Discussion
The discussion should mention that the pressure of the grooming procedures done by humans in this study was not measured and therefore not controlled and may also play a role in the outcomes of EAI horses versus RS horses. Although this is mentioned a little in the conclusion, it would be beneficial to include it here.
Author Response
Overall: Exceptionally well-written and clear with regards to literature review, methods, results, and discussion of the results in the context of existing studies. Very well done!
Abstract: Well-written and covers the hypothesis, materials and methods, findings, and conclusions with concise clarity.
Thank you
Line 28- a word is missing between “such” and “working”
L28: done: “such as working”
Introduction: Overall, very thorough and well-written, covering all important topics related to the experiment.
Thank you
Line 47 – 48: Consider mentioning that olfactory senses of environmental hormones are also involved in mate choice
L47-48: done: For example, mate choice necessitates first seeing hearing or smelling a conspecific and recognizing it as a potential mate [1].
Line 55-63: Consider adding in that differences in skin nerve receptors can also play a role.
Perceptual enhancement hypothesis is given by the cited studies (e.g. Ragert et al, 2004; Reuter et al., 2012). These studies do not mentioned any differences in skin nerve receptors according to individuals’ experience.
Materials and Methods: Overall very good and thorough in the explanation of the experiment and design.
Thank you
Line 193: Please clarify the “lived for at least one year”. Do you mean within the center or among familiar horses, etc?
L193: done: “within the center”
General Discussion
The discussion should mention that the pressure of the grooming procedures done by humans in this study was not measured and therefore not controlled and may also play a role in the outcomes of EAI horses versus RS horses. Although this is mentioned a little in the conclusion, it would be beneficial to include it here.
L485: done: “Further studies measuring brushing pressure and velocity may be of great interest.”